

Atmospheric
Measurement
Techniques

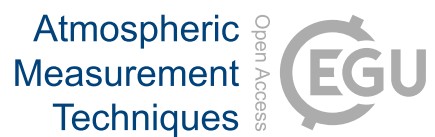

# Caution with spectroscopic NO₂ reference cells (cuvettes)

**Ulrich Platt**[1,2] **and Jonas Kuhn**[1,2]

[1]Institute of Environmental Physics (IUP), Heidelberg University, INF 229, 69120 Heidelberg, Germany
[2]Max Planck Institute for Chemistry, Mainz, Germany

**Correspondence:** Ulrich Platt (ulrich.platt@iup.uni-heidelberg.de)

**Abstract.** Spectroscopic measurements of atmospheric trace gases, for example, by differential optical absorption spectroscopy (DOAS), are frequently supported by recording the trace-gas column density (CD) in absorption cells (cuvettes), which are temporarily inserted into the light path. The idea is to verify the proper functioning of the instruments, to check the spectral registration (wavelength calibration and spectral resolution), and to perform some kind of calibration (absolute determination of trace-gas CDs). In addition, trace-gas absorption cells are a central component in gas correlation spectroscopy instruments. In principle DOAS applications do not require absorption-cell calibration; however, in practice, measurements with absorption cells in the spectrometer's light path are frequently performed.

Since NO₂ is a particularly popular molecule to be studied by DOAS, and at the same time it can be unstable in cells, we chose it as an example to demonstrate that the effective CD seen by the instrument can deviate greatly (by orders of magnitude) from expected values. Analytical calculations and kinetic model studies show the dominating influence of photolysis and dimerization of NO₂. In particular, this means that the partial pressure of NO₂ in the cell matters. However, problems can be particularly severe at high NO₂ pressures (around $10^5$ Pa) as well as low NO₂ partial pressures (of the order of a few 100 Pa). Also, it can be of importance whether the cell contains pure NO₂ or is topped up with air or oxygen (O₂). Some suggestions to improve the situation are discussed.

## 1 Introduction

There are a number of reasons for using absorption cells in conjunction with instruments measuring trace-gas column densities (CDs) by absorption spectroscopy, e.g. by differential optical absorption spectroscopy (DOAS). These include the verification of the overall functioning of the instrument, stray-light determination, or a check of the instrument's absolute wavelength calibration.

Field calibration of a spectrometer is not necessary for UV-visible absorption spectroscopy (see e.g. Platt and Stutz, 2008), since the instrument can be calibrated by using high-resolution absorption cross-section spectra of the particular gases. This is accomplished by (1) determining the instrument function (IF) and (2) convoluting a high-resolution trace-gas cross-section spectrum with this IF and then (3) fitting the resulting trace-gas cross section to measured spectra in order to obtain the trace-gas CD. The details of this process are explained in studies, e.g. by Platt and Stutz (2008). However, it may be tempting to perform the calibration process simply by recording the CD of an absorption cell filled with a known amount of trace gas brought into the light path of the instrument. This approach complicates the measurements and may introduce additional errors due to uncertainties in the trace-gas CD in the cell. Nevertheless such procedures may work for a series of gases, like O₂, CO, CO₂, and CH₄, which do not (at ambient temperature) undergo self-reaction and which are neither photolysed by ambient solar radiation near the Earth's surface nor by the radiation typically used for their absorption spectroscopic measurement.

On the other hand, if a trace-gas cell is used to determine absolute wavelength calibration of a spectrometer, the absolute trace-gas CD in the cell is usually not critical.

In addition, gas correlation spectroscopy measurements (e.g. Ward and Zwick, 1975; Sandsten et al., 1996, 2004; Kebabian et al., 2000) require absorption cells containing the gas to be measured at CDs leading to optical densities around unity.

In general, there are a number of issues with using gas cells for these purposes, including the following:

– optical problems with the cell

– stability of the gas in the cell due to photolysis and/or other chemical reactions

– temperature dependence of chemical equilibria within the cell

– temperature dependence of the optical density.

In the following we discuss the above problems for the case of $NO_2$-absorption cells; however some of the discussed issues will also apply to cells with other gases.

## 2   Optics of cells

In principle the introduction of an absorption cell into the optical path of a remote-sensing instrument (e.g. a spectrometer) is straightforward. The cell is mounted in front of the entrance optics, and in the first approximation the absorption due to the trace gas in the cell (i.e. due to the trace-gas CD) is added to the trace-gas absorption seen without the cell. While this view is correct in some approximation, in detail there are a number of problems that need investigation.

### 2.1   Path length in an isolated cell

In a realistic cell, partial reflection (reflectance $R$) occurs at the cell windows. For simplicity we assume an index of refraction of $n = 1.5$ for the cell window material and accordingly (Fresnel formula)

$$R = \left(\frac{n-1}{n+1}\right)^2 \approx 0.04, \tag{1}$$

i.e. about 4 % reflection per surface for near-normal incidence (see Fig. 1). The reduction of the incoming intensity by $(1-R)^4$, or about 15 %, is probably of minor importance; however (if we neglect the absorption by the trace gas in the cell) a fraction of about $(1-R)^2 \cdot (R+R)^2 \approx 0.59$ % of the incoming radiation and 0.69 % of the transmitted radiation passes the cell three times (this effect will be lower at high trace-gas optical densities and also could be reduced by adding anti-reflective coatings to the cell windows). Due to this multiply reflected light the total absorption of the cell (and thus the trace-gas slant column density – SCD – $S_C$) will be enhanced by about 2 % over the CD $S_0$ for a single traverse. We note that the case of nearly normal incidence is quite realistic in many cases; for instance multi-axis

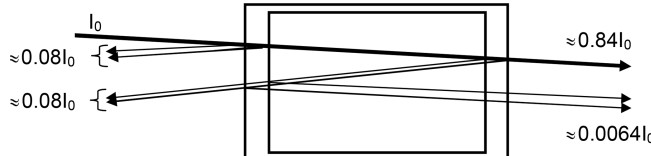

**Figure 1.** Sketch of the optics of a gas absorption cell; parallel rays are assumed, and the (small) tilt of the incoming ray with respect to the cell axis is introduced to distinguish the rays. The described effect will also be there at strictly normal incidence. We assume an index of refraction of $n = 1.5$ for the cell window material and accordingly 4 % reflection per surface (for near-normal incidence). Note that a fraction of about 0.69 % of the transmitted radiation passes the cell three times, thus adding $\approx 2$ % to the total absorption (if the trace-gas absorption in the cell is small).

DOAS CE1 (MAX-DOAS) instruments (e.g. Hönninger and Platt, 2002; Platt and Stutz, 2008) have total aperture angles of the order of $1°$, i.e. incidence angles of $-0.5$ to $+0.5°$. A typical use of a cell would be to mount it just in front of the instrument's telescope. In this arrangement the enhancement of the light path inside the cell (and thus the trace-gas column density $S_C$) due to the finite aperture angle of the radiation passing the cell will vary according to $S_C = S_0/\cos(\vartheta)$, with $S_0$ denoting the trace-gas CD for rays parallel to the cell axis. An angle of $\vartheta = 0.5°$ would lead to an enhancement in $S_C/S_0$ of $\approx 1.000038$ or 0.004 %.

Thus a slight (few degrees) tilt of the cell will not lead to noticeable light path extension in the cell, but already a $1°$ tilt, leading to 0.015 % light path extension, would be sufficient to direct the multiple reflected light outside the field of view of the telescope. Thus, the additional 2 % cell absorption would disappear. On the other hand, larger tilts of the cell, for example of $10°$, would increase the cell absorption again by 1.5 % and should therefore be avoided. This could be accomplished by a rigid mount which fixes the (removable) cell at a defined angle with respect to the cell optical axis (normal of the windows), e.g. at $2°$.

As will be discussed below, the acceptance angle of the cell to ambient direct or scattered sunlight can play a significant role. Therefore, this small aperture angle allows for shielding the cell from sunlight, since solar radiation only needs to enter from a small solid angle (of the order of $10^{-3}$ sr), for instance, by mounting the cell inside a relatively long tube made of nontransparent material.

### 2.2   Path length in a cell as part of an optical system

In Sect. 2.1 we discussed the behaviour of an isolated cell; however the idea is to incorporate an absorption cell into an optical system, i.e. to just hold it in front of a MAX-DOAS instrument. In this case there can be interaction between the cell and the entrance optics of the instrument (for instance due to reflection of light at the surface of the telescope lens). As described by Lübcke et al. (2013) this can further enhance

the trace-gas CD in the cell as seen by the instrument looking through it.

In the case of using gas cells in imaging instruments, for instance imaging spectrometers (e.g. Lohberger et al., 2004) or gas correlation instruments (e.g. Ward and Zwick, 1975), a larger aperture angle is required. This causes two potential problems. First, the aperture angle of the cell has to be much larger than in the case of a one-pixel (narrow field of view) instrument, for instance, typically a 30° total angle. Thus the acceptance angle for solar radiation becomes considerably larger (e.g. 0.22 sr instead of $10^{-3}$ sr), and consequently the photolysis frequencies for the gases inside the cell will be enhanced (see below). Second, the trace-gas CD of the cell becomes dependent on the observation angle $\vartheta$ (angle between the optical axis and the actual viewing direction within the field of view) as described above. For a total aperture angle of 30° this would amount to an enhancement of the $S_C(15°)$ over $S_0 = S_C(0°)$ of about 3.5 %.

## 3 Chemistry in NO$_2$-absorption cells

Nitrogen dioxide (NO$_2$) is a quite reactive gas (see rest of the section); therefore a series of chemical processes in an absorption cell can occur. Since they can alter the NO$_2$ concentration – and thus the NO$_2$ CD in the cell – considerably, they have to be watched. In the following subsections we discuss the relevant chemical processes, starting with important reactions and then proceeding to further reactions which are only relevant under certain conditions or if high accuracy is required.

We first discuss simplified chemistry, just encompassing the pertinent reactions, and then we proceed to a more comprehensive discussion of the chemistry in the following subsections.

### 3.1 The (initial) NO$_2$-only chemistry – simple case

In a cell (initially) filled only with NO$_2$ we can expect a series of reactions to occur, which are described in the following (bi-molecular rate constants are given in cm$^3$ molec.$^{-1}$ s$^{-1}$, termolecular rate constants are given in cm$^6$ molec.$^{-2}$ s$^{-1}$ for 25 °C and 1000 hPa, and details on temperature and pressure dependence as well as literature references can be found in Table 1). In fact, when buying NO$_2$ from a manufacturer, some of the described reactions can already proceed in the initial gas, which therefore might already contain impurities (e.g. of NO, HONO, and HNO$_3$).

Usually cells are exposed to sunlight or radiation needed for the measurement; thus NO$_2$ in the cell can be photolysed:

$$\text{NO}_2 + h\nu \rightarrow \text{NO} + \text{O}(^3\text{P}), \quad J_1 \approx 8 \times 10^{-3}\,\text{s}^{-1}. \quad \text{(R1)}$$

The above value for $J_1$ is reached in full sunshine around noontime (see e.g. Jones and Bayes, 1973 or Kraus and Hofzumahaus, 1998). Of course this figure (and in fact all

photolysis frequencies in the cell; see Table 1) is highly variable, depending on solar zenith angle (i.e. latitude, season, and time of day), cloudiness, atmospheric turbidity, and the shading situation at the measurement site. In the case of active DOAS systems the photolysis frequencies will depend on the intensity of the light source and on the fraction of the cell cross-section area covered by the light beam. Nevertheless, it is frequently seen that calibration cells are used in full sunshine without any shielding; moreover, as is shown below (see Sect. 4), the effects on the NO$_2$ chemistry are similar over a wide range of photolysis frequencies.

In the following, ground-state oxygen atoms O($^3$P) will be denoted by O. The threshold wavelength for Reaction (R1) is about 398 nm (e.g. Johnston and Graham, 1974; Burkholder et al., 2015); however, due to vibrational excitation of the ground-state molecule there is noticeable photolysis up to about 430 nm. If the cell is only illuminated with radiation of a wavelength longer than 430 nm, NO$_2$ will not photolyse and $J_1$ will be essentially zero.

Although it is only a small effect it is worth noting that the photolysis frequency inside a cell is not different from the value in the air surrounding the cell despite reflectance of the cell walls, as described by, for example, Bahe et al. (1979).

The oxygen atoms produced in Reaction (R1) can (1) recombine

$$\text{O} + \text{O} + M \rightarrow \text{O}_2 + M,$$
$$k_2(298\,\text{K}, 10^5\,\text{Pa}) \approx 2.5 \times 10^{-14}\,\text{cm}^3\,\text{molec.}^{-1}\,\text{s}^{-1}. \quad \text{(R2)}$$

However, this is a slow process, because the O-atom concentration will be very low (see Figs. 4 to 8). Alternatively, (2) O atoms may react with the wall where they predominantly recombine (see e.g. Cartry et al., 2000):

$$\text{O} + \text{O} \rightarrow \text{Wall} \rightarrow \text{O}_2. \quad \text{(R3)}$$

Also, (3) O atoms can react with NO$_2$ to form NO:

$$\text{O} + \text{NO}_2 \rightarrow \text{NO} + \text{O}_2,$$
$$k_4(298\,\text{K}) \approx 2.52 \times 10^{-12}\,\text{cm}^3\,\text{molec.}^{-1}\,\text{s}^{-1}. \quad \text{(R4)}$$

Further, (4) oxygen atoms also may react with NO to form NO$_2$:

$$\text{O} + \text{NO} + M \rightarrow \text{NO}_2 + M,$$
$$k_5(298\,\text{K}, 10^5\,\text{Pa}) \approx 2.2 \times 10^{-12}\,\text{cm}^3\,\text{molec.}^{-1}\,\text{s}^{-1}. \quad \text{(R5)}$$

The final possibility, (5) formation of NO$_3$ – as well as further reactions, will be addressed in Sect. 3.4 below.

In addition there is the termolecular reaction of the O$_2$ formed in Reactions (R4) or (R2) (or added to the cell filling) that oxidizes NO to NO$_2$:

$$2\text{NO} + \text{O}_2 \rightarrow 2\text{NO}_2,$$
$$k_6 \approx 1.95 \times 10^{-38}\,\text{cm}^6\,\text{molec.}^{-2}\,\text{s}^{-1}. \quad \text{(R6)}$$

In an attempt to obtain a first-order quantitative understanding of the processes in the cell, we just consider a pure NO$_2$ initial filling and Reactions (R1) (NO$_2$ photolysis), (R4) (O + NO$_2$), and (R6) (2NO + O$_2$).

From the combination of Reactions (R1) and (R4) we derive the rates of NO and O$_2$ formation under illumination,

$$P(\text{NO}) = \frac{d}{dt}[\text{NO}] \approx 2 \cdot P(\text{O}_2) \approx 2 \cdot [\text{NO}_2] \cdot J_1, \qquad (2)$$

which ultimately (i.e. in the stationary state) must equal the rate of NO destruction, $D(\text{NO})$ and NO$_2$ formation, and $P(\text{NO}_2)$ due to Reaction (R6):

$$D(\text{NO}) = -\frac{d}{dt}[\text{NO}] \approx P(\text{NO}_2) \approx 2 \cdot [\text{NO}]_S^2 \cdot [\text{O}_2] \cdot k_6. \quad (3)$$

Since $P(\text{O}_2) \approx 0.5 \cdot P(\text{NO})$ and the concentration of both species are zero initially, we have $[\text{NO}] \approx 2 \cdot [\text{O}_2]$. Substituting this relationship,

$$D(\text{NO}) = -\frac{d}{dt}[\text{NO}] \approx [\text{NO}]^3 \cdot k_6, \qquad (4)$$

and equating $P(\text{NO})$ with $D(\text{NO})$, we obtain

$$2[\text{NO}_2] \cdot J_1 \approx [\text{NO}]_S^3 \cdot k_6. \qquad (5)$$

Further substituting $[\text{NO}_2] \approx [\text{NO}_2]_0 - [\text{NO}]_S$,

$$[\text{NO}_2]_0 - [\text{NO}]_S \approx [\text{NO}]_S^3 \cdot \frac{k_6}{2J_1}, \qquad (6)$$

or

$$[\text{NO}_2]_0 \approx [\text{NO}]_S^3 \cdot \frac{k_6}{2J_1} + [\text{NO}]_S. \qquad (7)$$

This cubic equation can be solved for the stationary-state NO concentration $[\text{NO}]_S$ as a function of the initial $[\text{NO}_2]_0$ as given in Appendix A.

*Examples.* (1) As an example, and to obtain a first idea of what might be happening in the cell, we assume about 1 atm (1000 hPa) of pure NO$_2$ (initially); i.e. the initial NO$_2$ concentrations in the cell will be $[\text{NO}_2]_0 \approx 2.4 \times 10^{19}$ cm$^{-3}$ and the very simple chemical system just comprising Reactions (R1), (R4), and (R6). As we show below, the simplified reaction system – with the exception of the NO$_2$ dimer (N$_2$O$_4$) formation (see Sect. 3.2) – is quite adequate. Also, such a cell would have a peak optical density (at around 440 nm) of about 14 at 1 cm length but much lower at other wavelengths.

In the dark ($J_1 = 0$) nothing will happen, while in sunlight ($J_1 = 8 \times 10^{-3}$): NO + O formation will take place followed by Reaction (R4) of NO$_2$ with O. Thus the (initial) rate of NO formation $P(\text{NO})$ will be

$$P(\text{NO}) \approx 2[\text{NO}_2]_0 \cdot J_1 \approx 3.8 \times 10^{17} \, \text{cm}^{-3} \, \text{s}^{-1}. \qquad (8)$$

This will lead to an initial decay time $\tau_{\text{NO}_2} = [\text{NO}_2]_0 / P(\text{NO}) = 1/(2J_1) \approx 63$ s. The stationary-state NO concentration can be calculated according to

Eq. (7) and the solution given in Appendix A to be $[\text{NO}]_S \approx 2.57 \times 10^{18}$ molec. cm$^{-3}$, or about 10.7 % of the initial NO$_2$ level. In other words, the NO$_2$ concentration will be reduced to 89.3 % of its initial value $[\text{NO}_2]_0$. The corresponding NO rate of destruction will be

$$D(\text{NO}) \approx [\text{NO}]^3 \cdot k_6 \approx 3.32 \times 10^{17} \, \text{molec. cm}^{-3} \, \text{s}^{-1},$$

matching

$$P(\text{NO}) \approx 2[\text{NO}_2]_S \cdot J_1,$$

from NO$_2$ photolysis .

(2) We give a further example using about 1 hPa of pure NO$_2$ (initially) corresponding to $[\text{NO}_2]_0 \approx 2.4 \times 10^{16}$ cm$^{-3}$, and the same simple chemical system, just comprising Reactions (R1), (R4), and (R6), as above. Such a cell would have an initial differential optical density in the vicinity of 450 nm of about $2.4 \times 10^{-3}$ and would thus appear ideal to test the sensitivity of an NO$_2$ spectrometer.

In sunlight we have $D(\text{NO}_2) \approx 1.92 \times 10^{14}$ cm$^{-3}$ s$^{-1}$. In this case, the resulting stationary-state NO level becomes $[\text{NO}]_S \approx 2.4 \times 10^{16}$, or about 100 % of the initial NO$_2$. In other words after illumination the remaining NO$_2$ concentration and thus the NO$_2$ CD of the cell will only be a very small fraction of the expected value ($[\text{NO}_2]_S \cdot J_1 = D(\text{NO})$ or $[\text{NO}_2]_S = D(\text{NO})/J_1 \approx 1.7 \times 10^{13}$ cm$^{-3}$, i.e. < 0.1 % of the initial $[\text{NO}_2]$). After a short (of the order of 1 min) exposure to sunlight the NO$_2$ in the cell will practically vanish.

On the other hand, in this simplified calculation, the NO reconversion, $D(\text{NO})$, to NO$_2$ will be much slower than the initial photolysis:

$$D(\text{NO}) \approx 0.5 \cdot [\text{NO}]^3 \cdot k_6 \approx 1.35 \times 10^{11} \, \text{molec. cm}^{-3} \, \text{s}^{-1}.$$

*Recovery from illumination.* A further interesting question concerns the time for the chemical system to recover from a period of photolysis. Equation (4) gives the rate of NO destruction as a function of [NO]. In the case of example (1), above NO would decay with an initial rate of $D(\text{NO})/[\text{NO}] \approx 0.11$ s$^{-1}$ (ca. 11 % per second, suggesting a 9 s time constant for recovery). However, $D(\text{NO})$ varies with the third power of [NO]. When, for example, 90 % the NO is consumed (i.e. 1.4 % of [NO] is still left) the time constant would increase by a factor of 1000 to around 3 h.

In the case of example (2), the initial reconversion rate would only be $5.6 \times 10^{-6}$ s$^{-1}$ (or $\approx 49$ % per day), which would seem to imply a recovery time of somewhat more than 2 d. But again the dependence on the cube of the NO concentration means that the recovery time becomes much longer later on. For some model results, see Fig. 9.

## 3.2 The NO$_2$ ↔ N$_2$O$_4$ equilibrium

An additional problem in NO$_2$ cells – in particular if high NO$_2$ concentrations approaching 1000 hPa are used – is the

formation of the dimer N$_2$O$_4$ (see also Roscoe et al., 1993):

$$2NO_2 + M \rightarrow N_2O_4 + M,$$

$$k_7(298\,\text{K}) \cdot [M] \approx 3.3 \times 10^{-14}\,\text{cm}^3\,\text{molec.}^{-1}\,\text{s}^{-1}. \quad (\text{R7})$$

There is a thermal decay of the dimer,

$$N_2O_4 + M \rightarrow 2NO_2 + M,$$

$$k_8(298\,\text{K}) \times [M] \approx 1.47 \times 10^5\,\text{s}^{-1}, \quad (\text{R8})$$

leading to an equilibrium with the equilibrium constant (298K; from Atkinson et al., 2004),

$$K_{\text{Eq}} = \frac{k_\rightarrow}{k_\leftarrow} = \frac{[N_2O_4]}{[NO_2]^2} \approx 2.29 \times 10^{-19}\,\text{cm}^3\,\text{molec.}^{-1}. \quad (9)$$

Note that the time to attain the equilibrium is shorter than $1/k_8 \approx 7\,\mu\text{s}$ (at 298 K and 1000 hPa). Thus, one can assume that there is always equilibrium between NO$_2$ and N$_2$O$_4$. From this follows, for the [NO$_2$]/[N$_2$O$_4$] ratio,

$$\frac{1}{K_{\text{Eq}}[NO_2]} = \frac{[NO_2]}{[N_2O_4]} \ \text{ or } \ [N_2O_4] = K_{\text{Eq}} \cdot [NO_2]^2. \quad (10)$$

What is usually most interesting is the fraction of NO$_2$ of the total amount of NO$_2$ + N$_2$O$_4$ (i.e. pressure during filling) in the cell. The fraction is given by [NO$_Z$] = [NO$_2$] + [N$_2$O$_4$] and thus

$$\frac{[NO_2]}{[NO_Z]} = \frac{[NO_2]}{[N_2O_4] + [NO_2]} = \frac{[NO_2]}{K_{\text{Eq}} \cdot [NO_2]^2 + [NO_2]}$$

$$= \frac{1}{K_{\text{Eq}} \cdot [NO_2] + 1}, \quad (11)$$

which can be transformed into

$$K_{\text{Eq}} \cdot [NO_2] + 1 = \frac{[NO_Z]}{[NO_2]} \Rightarrow [NO_Z]$$

$$= K_{\text{Eq}} \cdot [NO_2]^2 + [NO_2], \quad (12)$$

and solved for [NO$_2$],

$$[NO_2]^2 + \frac{[NO_2]}{K_{\text{Eq}}} - \frac{[NO_Z]}{K_{\text{Eq}}} = 0, \quad (13)$$

with the only positive solution:

$$[NO_2]_1 = -\frac{1}{2K_{\text{Eq}}} + \sqrt{\frac{1}{4K_{\text{Eq}}^2} + \frac{[NO_Z]}{K_{\text{Eq}}}}, \quad (14)$$

or

$$[NO_2]_1 = \frac{1}{K_{\text{Eq}}}\left(-\frac{1}{2} + \sqrt{\frac{1}{4} + [NO_Z] \cdot K_{\text{Eq}}}\right). \quad (15)$$

The relationship between NO$_2$ and [NO$_2$]/[NO$_Z$] in the cell as a function of total [NO$_Z$] = [NO$_2$] + [N$_2$O$_4$] is shown in Fig. 2.

For example, $[M] = 2.4 \times 10^{19}$ (1000 hPa or ca. 1 atm of total pressure, at 298 K), resulting in [NO$_2$]$_1 \approx 8.29 \times 10^{18}$ molec. cm$^{-3}$ and [NO$_2$]$_1$/[NO$_Z$] $\approx 0.344$. Thus, filling a cell from an NO$_2$ reservoir (e.g. an NO$_2$ tank) to 1 atm of total pressure will lead to only 34 % of this pressure being present as NO$_2$ (see also Fig. 2).

At 100, 10, and 1 hPa (ca. 0.1, 0.01, and 0.001 atm) of NO$_2$ + N$_2$O$_4$, the corresponding figures for [NO$_2$]$_1$/[NO$_Z$] would be 0.717, 0.95, and 0.995, respectively. These figures are independent of an additional topping with air or oxygen to a full atmosphere of total pressure, as is described below. In other words, unless the NO$_2$ partial pressure is below around 10 Pa, the actual NO$_2$ partial pressure (and thus the concentration of NO$_2$) will be below expected levels by two-digit percentages.

A further problem associated with the NO$_2$–N$_2$O$_4$ equilibrium is the marked temperature dependence of the equilibrium constant. In the usual Arrhenius expression, it is given as

$$K_{\text{Eq}}(T) = A \cdot e^{\frac{B}{T}}, \quad (16)$$

with $A = 1.07 \times 10^{-28}$ cm$^{-3}$ molec.$^{-1}$ and $B = 6400$ K (see Table 1). The (relative) temperature dependence of $K_{\text{Eq}}$ is given by

$$\frac{1}{K_{\text{Eq}}(T)}\frac{d}{dT}\left(K_{\text{Eq}}(T)\right) = \frac{1}{K_{\text{Eq}}(T)}A \cdot e^{\frac{B}{T}}\frac{d}{dT}\left(\frac{B}{T}\right)$$

$$= \frac{1}{K_{\text{Eq}}(T)} \cdot -\frac{AB}{T^2} \cdot e^{\frac{B}{T}} = -\frac{B}{T^2}, \quad (17)$$

and with the above values for $A$ and $B$, we obtain, for the relative change in the equilibrium constant,

$$\frac{1}{K_{\text{Eq}}(T)}\frac{d}{dT}\left(K_{\text{Eq}}(T)\right) = -\frac{B}{T^2} \approx -0.072\frac{1}{K}. \quad (18)$$

In other words the equilibrium constant is reduced by more than 7 % K$^{-1}$ of heating. Fortunately the effect on NO$_2$ is somewhat smaller, ranging from nearly zero change at very small NO$_2$ levels to about a 3 % increase per degree of heating at 1000 hPa (see Appendix B).

### 3.3 NO$_2$ + O$_2$ chemistry

The addition of O$_2$ (or air) to the NO$_2$ filling can greatly help with stabilizing the NO$_2$ concentration in a cell under certain conditions.

In the presence of molecular oxygen, following the photolysis of NO$_2$, ozone is formed in the cell:

$$O + O_2 + M \rightarrow O_3 + M,$$

$$k_9(298\,\text{K}) \approx 1.46 \times 10^{-14}\,\text{cm}^3\,\text{molec.}^{-1}\,\text{s}^{-1}. \quad (\text{R9})$$

This in turn can react with NO to form NO$_2$:

$$O_3 + NO \rightarrow NO_2 + O_2,$$

$$k_{10}(298\,\text{K}) \approx 1.9 \times 10^{-14}\,\text{cm}^3\,\text{molec.}^{-1}\,\text{s}^{-1}. \quad (\text{R10})$$

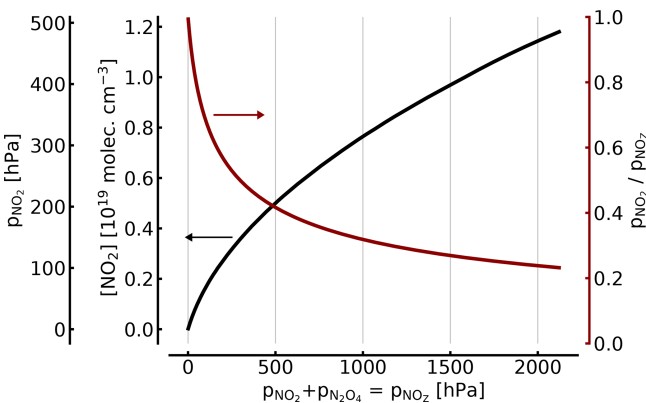

**Figure 2.** CE2 NO$_2$ concentration (black line in units of $10^{19}$ molec. cm$^{-3}$ and hPa; left axes) and fraction of NO$_2$ (red line; right axis) of the total $[NO_Z] = [NO_2] + [N_2O_4]$ as a function of $[NO_Z]$ (given in pressure units for 25 °C). At atmospheric pressure (1000 hPa) in the cell, only about 34 % of the total NO$_Z$ TS1 (or $\approx 344$ hPa partial pressure) exists as NO$_2$.

(a)

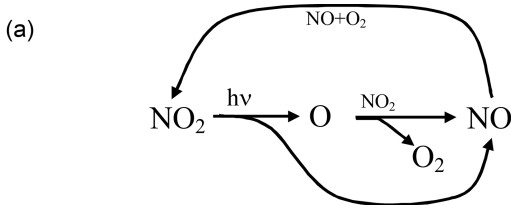

(b)

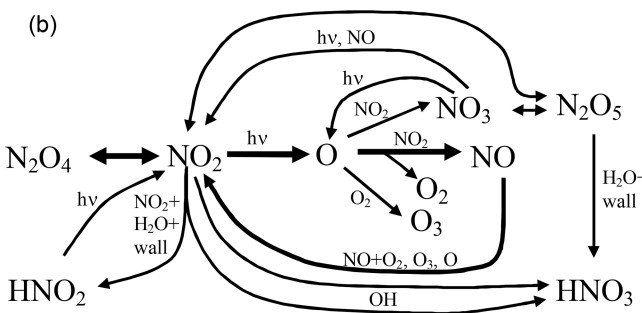

**Figure 3.** Scheme of the chemical reactions in an illuminated NO$_2$ cell; **(a)** only basic reactions and **(b)** complete system excluding the formation of OH.

The reaction scheme encompassing the reaction pathways discussed above is sketched in Fig. 3.

*Here we can distinguish two regimes.* The first regime assumes comparable concentrations of O$_2$ and NO$_2$, i.e. $[O_2]/[NO_2]$ around unity. In this case the termolecular oxidation of NO by O$_2$ dominates. This is similar to the situation discussed in Sect. 3.1; however we can take the O$_2$ concentration $[O_2]$ to be essentially constant. This reduces the third-order kinetics of Eq. (7) to pseudo-second-order or second-order kinetics, and we obtain

$$[NO] = \sqrt{\frac{P_{NO}}{2[O_2]k_{NO}}} \underset{P_{NO} \text{ substituted}}{\approx} \sqrt{\frac{[NO_2] \cdot J}{[O_2]k_{NO}}}$$

$$\frac{[NO]}{[NO_2]} \approx \sqrt{\frac{J}{[O_2][NO_2]k_{NO}}}. \qquad (19)$$

For example we may assume 0.5 atm (500 hPa) each of pure NO$_2$ and O$_2$ (initially); i.e. the initial concentrations of either species in the cell will be $[O_2]_0 = [NO_2]_0 \approx 1.2 \times 10^{19}$ cm$^{-3}$. In sunlight we have NO$_2$ photolysis (Reaction R1) followed by O + NO$_2$ (Reaction R4) plus oxidation of NO by O$_2$;

$$D(NO) \approx 2[NO]^2 \cdot [O_2]k_{NO} = P(NO).$$

From this stationary-state assumption we can calculate $[NO]_s$:

$$[NO]_S \approx \sqrt{\frac{P_{NO}}{2[O_2]k_{NO}}} \approx 0.054 \cdot [NO_2].$$

Thus, the NO$_2$ concentration would be reduced by only 5.4 % from its initial value once the cell is exposed to sunlight.

The second regime assumes a high $[O_2]/[NO_2]$ ratio (for instance larger than $10^4$) so that the reaction of O atoms formed in NO$_2$ photolysis is much more likely to react with O$_2$ than with NO$_2$. In this case for each molecule of NO$_2$ photolysed nearly one molecule of O$_3$ is formed, which will react with the NO molecule produced in the NO$_2$ photolysis. The O$_3$ concentration will rise until its reaction with NO balances the rate of NO$_2$ photolysis:

$$[NO][O_3]k_{10} = [NO_2] \cdot J_1. \qquad (20)$$

Since $[NO] \approx [O_3]$ we obtain

$$[NO]^2 k_{10} \approx [NO_2] \cdot J \Rightarrow [NO] \approx \sqrt{\frac{[NO_2] \cdot J}{k_{10}}}. \qquad (21)$$

For instance at $[NO_2] = 2.4 \times 10^{15}$ cm$^{-3}$ and about 1 atm (1000 hPa) of O$_2$, the stationary-state NO level would be $[NO] \approx 4.4 \times 10^{13}$ cm$^{-3}$, or about 1.8 % of the initial NO$_2$ concentration. Note that a small fraction (about $10^{-4}$ in this example) of the O atoms produced in the NO$_2$ photolysis would still react with NO$_2$ and form NO without a corresponding O$_3$ production (rate about $2 \times 10^9$ cm$^{-3}$ s$^{-1}$); thus the NO fraction in the cell would slowly grow until Reaction (R6) balances this process. At the above NO level the rate of NO$_2$ formation would be around $10^9$ cm$^{-3}$ s$^{-1}$; thus the NO level would slightly grow (by about 50 %) during several days of continuous illumination of the cell.

### 3.4 The (initial) $NO_2$-only chemistry – some complications

In addition to the three reactions described above, O atoms can recombine with $NO_2$ to form nitrate radicals, $NO_3$:

$$O + NO_2 + M \rightarrow NO_3 + M,$$
$$k_{11}(298\,K, 1\,atm) \approx 2.5 \times 10^{-11}\,cm^3\,molec.^{-1}\,s^{-1}.$$

(R11)

The $NO_3$ radicals formed in Reaction (R11) can be photolysed:

$$NO_3 + h\nu \rightarrow NO_2 + O, \quad J_{12a} \approx 0.19\,s^{-1}, \tag{R12a}$$
$$NO_3 + h\nu \rightarrow NO + O_2, \quad J_{12b} \approx 0.016\,s^{-1}. \tag{R12b}$$

The threshold wavelength is much longer than in the case of $NO_2$ ($J_1$), and the photolysis is much faster. Alternatively, $NO_3$ may react with NO (from Reaction R1) to re-form $NO_2$,

$$NO_3 + NO \rightarrow 2NO_2,$$
$$k_{13}(298\,K) \approx 2.6 \times 10^{-11}\,cm^3\,molec.^{-1}\,s^{-1}, \tag{R13}$$

or undergo self-reaction,

$$NO_3 + NO_3 \rightarrow 2NO_2 + O_2,$$
$$k_{14}(298\,K) \approx 2.3 \times 10^{-16}\,cm^3\,molec.^{-1}\,s^{-1}. \tag{R14}$$

Finally, and typically most likely, $NO_3$ will react with $NO_2$ to form dinitrogen pentoxide, $N_2O_5$:

$$NO_3 + NO_2 + M \rightarrow N_2O_5 + M,$$
$$k_{15}(298\,K) \approx 1.34 \times 10^{-12}\,cm^3\,molec.^{-1}\,s^{-1}. \tag{R15}$$

Dinitrogen pentoxide is thermally unstable and decays:

$$N_2O_5 + M \rightarrow NO_3 + NO_2 + M,$$
$$k_{16}(298\,K) \approx 2.98\,cm^3\,molec.^{-1}\,s^{-1}. \tag{R16}$$

In the absence of water (dry system) $N_2O_5$ will just be another reservoir potentially sequestering some of the $NO_2$. On the other hand, $N_2O_5$ is the anhydride of nitric acid and may react with water to form $HNO_3$. While the reaction of $N_2O_5$ plus water vapour appears to be exceedingly slow in the gas phase, it may react with a layer at the cell surface; details are given in Sect. 3.5.

Analysing the above system of reactions, one notices that loss of O atoms other than by Reactions (R4) or (R11) are of minor importance. This is underlined by the results of the model calculations using the full chemical system (see Table 1) presented in Sect. 4.

Therefore, we can summarize that each photolysis Reaction (R1) is followed by a conversion of $NO_2$ to NO (Reaction R4) or to $NO_3$ (Reaction R11). However, $NO_3$ is largely converted back to $NO_2$ by Reactions (R12a), (R13), and (to a minor extent) (R14); thus, in effect each photolysis act of $NO_2$ leads to the loss of approximately two $NO_2$ molecules. Essentially $NO_2$ would be converted to $NO + O_2$. In bright sunshine with $J_1 \approx 8 \times 10^{-3}\,s^{-1}$, this would lead to an $NO_2$ lifetime in the cell of $\tau(NO_2) \approx 1/(2 \cdot J_1) \approx 63\,s$, or roughly 1 min. Even if the cell is kept in the shade or is only exposed to indoor illumination where $J_1$ could be estimated to be 10 times (shade) to 100 times (indoor) smaller than in bright sunshine the conversion could be expected to proceed within around 10 min (shade) to 100 min (indoor) or even faster.

### 3.5 $NO_2 + O_2 +$ (trace) $H_2O$ chemistry

Since water is by far the most abundant (typical mixing ratios around 1 %) reactive trace gas in the ambient atmosphere (not counting nobel gases, $CO_2$, $H_2$, or $N_2O$), it may be possible that if traces of water enter the cell when it is filled, then a series of additional reactions may play a role (see e.g. Bahe et al., 1979):

$$O_3 + h\nu \rightarrow O(^1D) + O_2, \quad J_{17} \approx 3 \times 10^{-5}\,s^{-1}. \tag{R17}$$

Followed by quenching of $O(^1D)$ to $O(^3P)$ or the formation of hydroxyl (OH) radicals,

$$O(^1D) + H_2O \rightarrow 2OH,$$
$$k_{18}(298\,K) \approx 2.0 \times 10^{-10}\,cm^3\,molec.^{-1}\,s^{-1}. \tag{R18}$$

In an $NO_2$ cell, OH radicals are most likely to react with $NO_2$ (or NO) to form nitric acid (or nitrous acid; see below):

$$OH + NO_2 + M \rightarrow HNO_3 + M,$$
$$k_{19}(298\,K) \approx 1.05 \times 10^{-11}\,cm^3\,molec.^{-1}\,s^{-1}. \tag{R19}$$

Nitric acid is photolysed very slowly, and also its reaction with OH (to form $NO_3$) is slow; thus it will constitute a final sink of $NO_2$ (and water) in the cell. Alternatively, OH may react with NO to form nitrous acid:

$$OH + NO + M \rightarrow HNO_2 + M,$$
$$k_{20}(298\,K, 10^5\,Pa) \approx 9.7 \times 10^{-12}\,cm^3\,molec.^{-1}\,s^{-1}, \tag{R20}$$

which – in turn – is lost by photolysis,

$$HNO_2 + h\nu \rightarrow OH + NO,$$
$$J_{21}(298\,K) \approx 1.34 \times 10^{-3}\,s^{-1}. \tag{R21}$$

In addition, $N_2O_5$, formed in Reaction (R15), can react with (liquid) water adsorbed at the wall of the cell, also forming $HNO_3$:

$$N_2O_5 + (H_2O)_{liq} \rightarrow 2(HNO_3)_{liq}. \tag{R22}$$

Finally $NO_2$ is also known to heterogeneously react with water:

$$NO_2 + NO_2 + (H_2O)_{liq} \rightarrow HNO_2 + (HNO_3)_{liq}. \qquad (R23)$$

Although this reaction appears to be second order in $NO_2$, several studies (e.g. by Kleffmann et al., 1998) found a first-order dependence of HONO formation on the $NO_2$ concentration probably because the $NO_2$ reaction with $NO_2$ adsorbed at the wall is rate limiting. Therefore, heterogeneous reactions of $N_2O_4$ with water are probably not important. $HNO_2$ will photolyse relatively quickly to form $OH + NO$ (with OH in most cases reacting according to Reaction R19); and $HNO_3$ from the above two reactions will remain. Our model calculations actually show that all $H_2O$ is ultimately (typically after a few hours in full sunshine) converted to $HNO_3$, sequestering equivalent amounts of $NO_2$ and water; thus no HONO will remain after this time.

For example a cell having been filled with a very small amount of $NO_2$ (e.g. 10 hPa or $2.4 \times 10^{17}$ molec. cm⁻³) is topped with ambient (e.g. laboratory) air (which is of course not recommended; see Sect. 5.2) at 25 °C and 70 % relative humidity. Thus the approximate amount of water admitted is 70 % of the saturation vapour pressure of $H_2O$ at that temperature (70 % of 31.6 hPa = 22.1 hPa or $5.3 \times 10^{17}$ molec cm⁻³). Some of this water will form a film at the inside of the cell and allow heterogeneous Reactions (R22) and (R23), converting $NO_2$ into $HNO_3$, although it is hard to judge how fast this process will proceed. In addition, upon illumination with UV-radiation Reactions (R9), (R17), (R18), and (R19) will provide (relatively slow) gas-phase conversion of $NO_2$ to $HNO_3$. Since the amount of $H_2O$ in this example exceeds the amount of $NO_2$ it is likely that ultimately all $NO_2$ is converted to $HNO_3$, as can be seen in Figs. 4 to 8.

## 4 Gas kinetic simulations

We performed a series of gas-kinetic-simulation calculations in order to illustrate the behaviour of the reaction system described above under various conditions of initial $NO_2$ and amounts of added $O_2$. In a one-box model, the system of coupled ordinary differential equations resulting from the above reactions was solved numerically. This allows following the temporal evolution of the concentration of the individual gases in the cell under given conditions. Our full model includes Reactions (R1) to (R23), except (R3) and (R22) of Table 1 (marked with an asterisk in column 1). The heterogeneous Reaction (R23) was included in the simulation, with the parameterization proposed by Kleffmann et al. (1998) $k_{23} = 0.25 \cdot S/V \cdot v \cdot \gamma$, with uptake coefficient $\gamma = 10^{-6}$, the molecular velocity of $NO_2$ $v$, and the surface-to-volume ratio $S/V$). We assumed a typical cell (cylindrical, radius of 1 cm, and length of 5 cm, $S/V = 240\,\text{m}^{-1}$) as well as estimates of the amount of $H_2O$ in this cell from assuming a monolayer of water at the inner surface of the cuvette. Of course – given

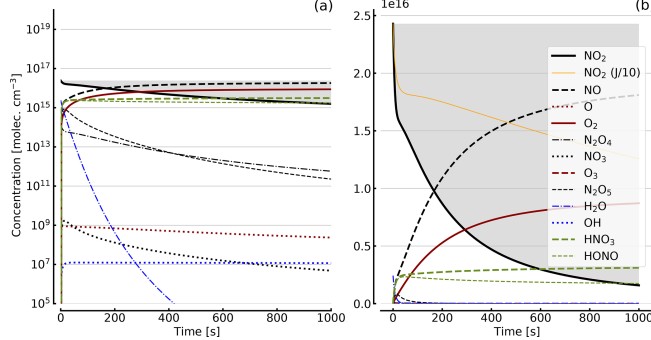

**Figure 4.** Results of calculations with the full model (reactions marked with * in Table 1). Shown are the temporal evolutions of [$NO_2$], [$NO_2$] with $J$ values scaled to 1/10, [NO], [O], [$O_2$], [$O_3$], [$N_2O_4$], [$NO_3$], [$N_2O_5$], [$H_2O$], [OH], [$HNO_3$], and [HONO] in an illuminated $NO_2$ cell. Initial [$NO_2$]₀ = 1 hPa ($2.4 \times 10^{16}$ molec. cm⁻³). Zero initial $O_2$ concentration was assumed; **(a)** logarithmic scale and **(b)** linear scale.

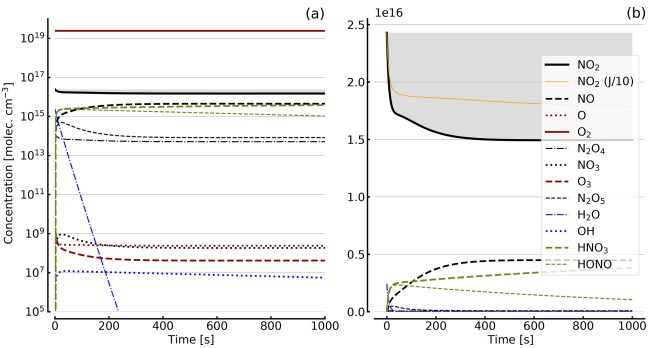

**Figure 5.** Results of calculations with the full model (reactions marked with * in Table 1). Same as Fig. 4, but with initial $O_2$ assumed; **(a)** logarithmic scale and **(b)** linear scale.

the uncertainties in heterogeneous reactions – this approach can only provide a rough estimate of the HONO concentration in the cell.

Also some runs with a subset of the reactions were performed as described in Appendix B in order to check on the simplified analytical calculations in the previous section. Table 2 shows a comparison of the $NO_2$ reduction after 1000 s given by the analytical calculations and the simplified model.

Further calculations encompass the full range of Reactions (R1) to (R23), except (R3) and (R22) as given in Table 1 (marked with an asterisk), where an analytical solution is not practical or is probably even impossible. Figures 4 to 8 show the results of these model runs for $NO_2$, NO, O atoms, $O_2$, $N_2O_4$, $NO_3$, $O_3$, $N_2O_5$, $H_2O$, OH, $HNO_3$, and HONO in an illuminated $NO_2$ cell for initial, $N_2O_4$-equilibrated [$NO_2$]₀ of 1, 10, 71, and 344 hPa ($2.4 \times 10^{16}$, $2.4 \times 10^{17}$, $1.7 \times 10^{18}$, and $0.84 \times 10^{19}$ molec. cm⁻³). In Fig. 4 no initial $O_2$ was assumed, and the remaining figures (Figs. 5 to 8) show time series with initial $O_2$. The left and right panels have logarithmic

**Table 1.** Summary of reaction rate constants TS2.

| No. | Reaction | $k(T)$, $J$, or $k_0(T)$ (in cm$^3$ molec.$^{-1}$ s$^{-1}$, if not given otherwise) | $k_\infty$ | $k$(298 K, 1 atm) or $J$ (in cm$^3$ molec.$^{-1}$ s$^{-1}$, if not given otherwise) |
|---|---|---|---|---|
| (1)[d],* | $NO_2 + h\nu \rightarrow NO + O$ | $8 \times 10^{-3}$ s$^{-1}$ | | $8 \times 10^{-3}$ s$^{-1}$ |
| (2)[c],* | $O + O + M \rightarrow O_2 + M$ | $5.21 \times 10^{-35} e^{(900/T)}$ cm$^6$ molec.$^{-2}$ s$^{-1}$ | | $2.51 \times 10^{-14}$ |
| (3) | $O + O \rightarrow Wall \rightarrow O_2$ | Neglected | | |
| (4)[a],* | $O + NO_2 \rightarrow NO + O_2$ | $5.1 \times 10^{-12} e^{(-210/T)}$ | | $2.52 \times 10^{-12}$ |
| (5)[a],* | $O + NO + M \rightarrow NO_2 + M$ | $9 \times 10^{-32} (T/300)^{-1.5}$ cm$^6$ molec.$^{-2}$ s$^{-1}$ | $3.0 \times 10^{-11}$ | $2.2 \times 10^{-12}$ |
| (6)[b],* | $2NO + O_2 \rightarrow 2NO_2$ | $3.3 \times 10^{39} \exp(530/T)$ cm$^6$ molec.$^{-2}$ s$^{-1}$ | | $1.95 \times 10^{-38}$ cm$^6$ molec.$^2$ s$^{-1}$ |
| (7)[b],* | $2NO_2 + M \rightarrow N_2O_4 + M$ | $1.4 \times 10^{-33} (T/300)^{-3.8}$ cm$^6$ molec.$^{-2}$ s$^{-1}$ | $1.0 \times 10^{-12}$ | $3.3 \times 10^{-14}$ |
| (8)[b],* | $N_2O_4 + M \rightarrow 2NO_2 + M$ | $1.3 \times 10^{-5} (T/300)^{-3.8} e^{(-6400/T)}$ | $1.15 \times 10^{16} e^{(-6460/T)}$ s$^{-1}$ | $1.47 \times 10^5$ s$^{-1}$ |
| (9)[a],* | $O + O_2 + M \rightarrow O_3 + M$ | $6.0 \times 10^{-34} (T/300)^{-2.4}$ cm$^6$ molec.$^{-2}$ s$^{-1}$ | $k_0[M] \ll k_\infty$ at 1000 hPa | $1.46 \times 10^{-14}$ |
| (10)[a],* | $O_3 + NO \rightarrow NO_2 + O_2$ | $3.0 \times 10^{-12} e^{(1500/T)}$ | | $1.9 \times 10^{-14}$ |
| (11)[a],* | $O + NO_2 + M \rightarrow NO_3 + M$ | $2.5 \times 10^{-31} (T/300)^{-1.8}$ | $2.2 \times 10^{-11} (T/300)^{-0.7}$ | $6.1 \times 10^{-12}$ |
| (12a)[b],* | $NO_3 + h\nu \rightarrow NO_2 + O$ | $0.19$ s$^{-1}$ | | $0.19$ s$^{-1}$ |
| (12b)[b],* | $NO_3 + h\nu \rightarrow NO + O_2$ | $0.016$ s$^{-1}$ | | $0.016$ s$^{-1}$ |
| (13)[a],* | $NO_3 + NO \rightarrow 2NO_2$ | $1.5(10^{-11} e^{(170/T)}$ | | $2.6 \times 10^{-11}$ |
| (14)[a],* | $NO_3 + NO_3 \rightarrow 2NO_2 + O_2$ | $8.5 \times 10^{-13} e^{(-2450/T)}$ | | $2.3 \times 10^{-16}$ |
| (15)[a],* | $NO_3 + NO_2 + M \rightarrow N_2O_5 + M$ | $2.4 \times 10^{-30} (T/300)^{-3.0}$ cm$^6$ molec.$^{-2}$ s$^{-1}$ | $1.6 \times 10^{-12} (T/300)^{0.1}$ | $1.34 \times 10^{-12}$ |
| (16)[b],* | $N_2O_5 + M \rightarrow NO_3 + NO_2 + M$ | $1.3 \times 10^{-3} (T/300)^{-3.5} e^{(-11\,000/T)}$ s$^{-1}$ | $9.7 \times 10^{14} (T/300)^{0.1} e^{(-11\,080/T)}$ s$^{-1}$ | $2.98$ TS3 s$^{-1}$ |
| (17)[e],* | $O_3 + h\nu \rightarrow O(^1D) + O_2$ | $3 \times 10^{-5}$ s$^{-1}$ | | $3 \times 10^{-5}$ s$^{-1}$ |
| (18)[a],* | $O(^1D) + H_2O \rightarrow 2OH$ | $1.63 \times 10^{-10} \times (e^{(60/T)})$ | | $2.0 \times 10^{-10}$ |
| (19)[a],* | $OH + NO_2 + M \rightarrow HNO_3 + M$ | $1.8 \times 10^{-30} (T/300)^{-3}$ | $2.8 \times 10^{-11}$ | $1.05 \times 10^{-11}$ |
| (20)* | $OH + NO + M \rightarrow HNO_2 + M$ | $7 \times 10^{-31} (T/300)^{-2.6}$ | $3.6 \times 10^{-11} (T/300)^{-0.1}$ | $9.7 \times 10^{-12}$ |
| (21)[f],* | $HNO_2 + h\nu \rightarrow OH + NO$ | $1.34 \times 10^{-3}$ | | $1.34 \times 10^{-3}$ |
| (22) | $N_2O_5 + (H_2O)_{liq} \rightarrow 2(HNO_3)_{liq}$ | Neglected | | |
| (23)* | $NO_2 + NO_2 + H_2O \rightarrow HNO_2 + HNO_3$ | See text | | |

[a] Data from Burkholder et al. (2015) *JPL Publication No. 15–10*. [b] Data from Atkinson et al. (2004). [c] Data from Tsang and Hampson (1986). [d] Data from Trebs et al. (2009). [e] Data from Bahe and Schurath (1978). [f] Data from Alicke et al. (2002). The reactions marked with * are included in the kinetic model; see Sect. 4.

**Table 2.** Comparison of the analytical calculations and the simplified model encompassing Reactions (R1), (R4), and (R6).

| Initial NO$_2$ cell pressure | NO$_2$ reduction after 1000 s (%) | | | |
|---|---|---|---|---|
| | Without O$_2$ | | Topped with O$_2$ | |
| | Simple model | Full model | Simple model | Full model |
| 1 hPa | 100 | 93 | 55 | 38 |
| 10 hPa | 93 | 75 | 20 | 17 |
| 71 hPa | 24 | 17 | 4 | 10 |
| 344 hPa | 3 | 3 | | |

and linear concentration scales, respectively. Comparison of the result with the data in Figs. 5 to 8 shows that there are no fundamental differences in the NO$_2$ time series between the simple model and the full model (see also Table 2).

In order to study the effect of different photolysis frequencies, we also performed model runs with all $J$ values scaled to 1/10 of the figures given in Table 1. The resulting temporal evolutions of NO$_2$ are also included in Figs. 4 to 8 (note that the time series of all other species are for the $J$ values as given in Table 1). As can be seen from these figures there is still a rather large loss of NO$_2$, even with $J$ being only 1/10 of its value in full sunshine. This is due to the fact that

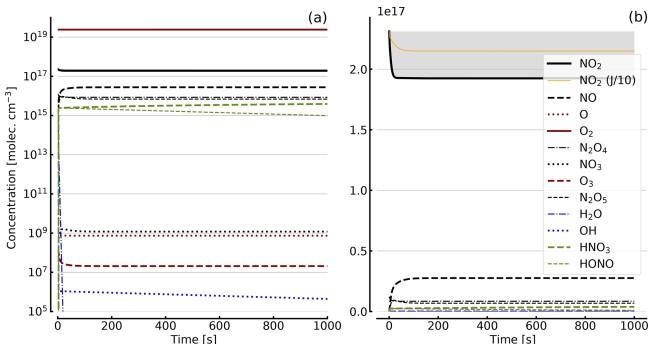

**Figure 6.** Results of calculations with the full model (reactions marked with * in Table 1). Same as Fig. 5, but initial $[NO_2]_0 = 10$ hPa ($2.4 \times 10^{17}$ molec. cm$^{-3}$) with initial $O_2$ assumed; **(a)** logarithmic scale and **(b)** linear scale.

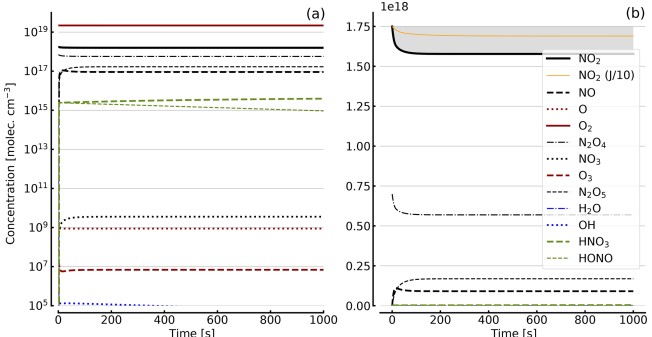

**Figure 7.** Results of calculations with the full model (reactions marked with * in Table 1). Same as Fig. 5, but initial $[NO_2]_0 = 71$ hPa ($1.7 \times 10^{18}$ molec. cm$^{-3}$) with initial $O_2$ assumed; **(a)** logarithmic scale and **(b)** linear scale.

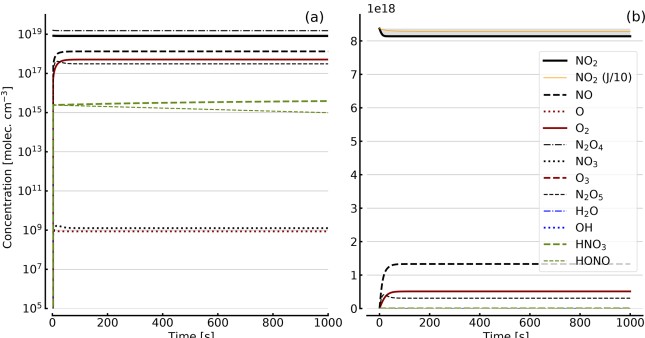

**Figure 8.** Results of calculations with the full model (reactions marked with * in Table 1). Same as Fig. 5, but initial $[NO_2]_0 = 344$ hPa ($0.84 \times 10^{19}$ molec. cm$^{-3}$) with initial $O_2$ assumed; **(a)** logarithmic scale and **(b)** linear scale.

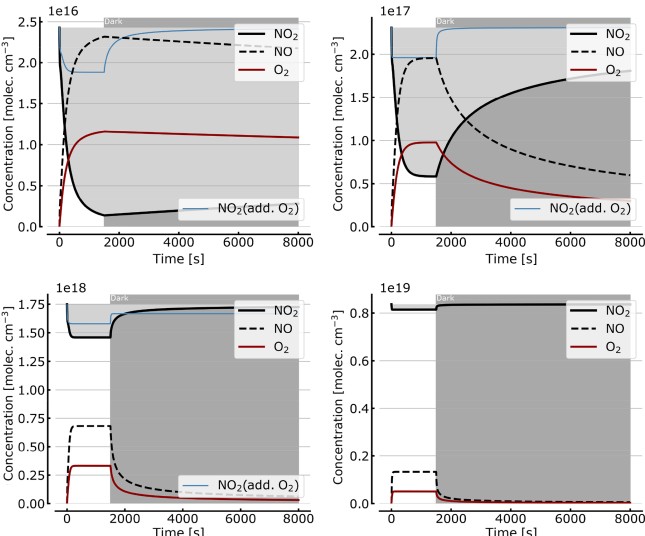

**Figure 9.** Recovery of NO₂ in the dark after initial illumination. Model calculations of the temporal evolution of $[NO_2]$ (thick solid black line), $[NO]$ (dashed black line), and $[O_2]$ (solid brown line) calculated with the full model (reactions marked with * in Table 1). The NO₂ cell is initially illuminated for 1500 s and then left in the dark afterwards. Initial $[NO_2]_0$ of 1, 10, 71, and 344 hPa ($2.4 \times 10^{16}$, $2.4 \times 10^{17}$, $1.7 \times 10^{18}$, and $0.84 \times 10^{19}$ molec. cm$^{-3}$). The blue thin line in the plots for 1, 10, and 71 hPa show $[NO_2]$ for $O_2$-topped-up cell.

the NO₂ loss scales only with the second or third root of the photolysis frequency (see analytical solutions in Sect. 3 and Appendix A).

As discussed in Sect. 3.1 the recovery of NO₂ in the dark after initial illumination (e.g. due to a use of the cell in a measurement) is an important question. Figure 9 shows model calculations of the temporal evolution of NO₂, NO, and O₂ according to the full model (Reactions R1 to R23, except Reactions R3 and R22 – see Table 1; at 298 K). The NO₂ cell is initially illuminated for 1500 s and then left in the dark afterwards for initial, N₂O₄-equilibrated $[NO_2]_0$ of 1, 10, 71, and 344 hPa ($2.4 \times 10^{16}$, $2.4 \times 10^{17}$, $1.7 \times 10^{18}$, and $0.84 \times 10^{19}$ molec. cm$^{-3}$, respectively). At the two highest $[NO_2]_0$ levels the initial NO₂ was chosen such that total pressures of 100 and 1000 hPa were reached. It can be seen that the NO₂ recovery at low NO₂ levels can take days to hours. Adding O₂ to the cell again has a strong impact on the $[NO_2]$ evolution (see thin blue lines in Fig. 7), reducing the recovery time to a fraction of the NO₂-only case. For larger initial NO₂ concentrations (e.g. 71 hPa) and added O₂, a hysteresis between initial $[NO_2]$ and equilibrium $[NO_2]$ in the dark

can be observed; i.e. the NO₂ level does not return to its initial value after illumination. This is due to the formation of N₂O₅ in the illuminated period.

## 5 Summary and conclusions

We conclude that the use of NO₂ cells requires careful consideration, in particular when quantitative measurements of the NO₂ CD in the cell are desired. If unfortunate parameters are chosen (e.g. rather low NO₂ pressures or no O₂ or air

added), practically no NO$_2$ might be found in the cell at all. Also, one cannot conclude that particularly high or low NO$_2$ concentrations in the cell are the superior choice. At high NO$_2$ concentrations (approaching atmospheric pressure) a large fraction of the NO$_2$ is converted to the dimer N$_2$O$_4$, which not only reduces the NO$_2$ CD way below expected values but also introduces a large temperature dependence (up to 3 % per degree) of the NO$_2$ CD in the cell (also, there might be some additional uncertainty due to uncertainty of the equilibrium constant, as pointed out by Roscoe and Hind, 1993). On the other hand, at low NO$_2$ levels (e.g. 1 hPa) photolysis may convert much (if not virtually all) of the NO$_2$ to NO. Although NO$_2$ eventually recovers, this process may take long (days) to complete. Thus, the actual NO$_2$ CD of the cell may become dependent on the illumination and recovery history of the cell and may be rather unpredictable for a particular cell.

Unfortunately, the two described effects are not even the full story; therefore the potential problems are listed below. Fortunately, there are ways to minimize the problems, like oxygen addition to the cell and choosing the right NO$_2$ concentration, which may help in reducing the uncertainty of the NO$_2$ CD of a given cell to the single-digit percentage range.

## 5.1 Summary of problems

As discussed above, the NO$_2$ concentration in a cell – and thus the NO$_2$ CD of the cell – can deviate from expectations due to a number of reasons:

1. Optical effects, namely multiple reflection in the cell and tilt of the cell with respect to the optical axis, can enhance the light path and thus the apparent NO$_2$ CD.

2. Photolysis of NO$_2$ can reduce the NO$_2$ CD in the cell.

3. Sequestration of NO$_2$ as N$_2$O$_4$ due to the thermodynamic equilibrium between the two species can reduce NO$_2$ in the cell and cause temperature dependence of the NO$_2$ CD.

4. Formation or re-formation of NO$_2$ from NO in the cell leads to slow recovery of NO$_2$.

5. Irreversible conversion or conversion of NO$_2$ to HNO$_3$ can lead to long-term loss of NO$_2$.

6. Wall loss of NO$_X$ species like N$_2$O$_4$ or N$_2$O$_5$ can lead to long-term loss of NO$_2$

## 5.2 Some ideas to remedy the situation

One approach for minimizing loss of NO$_2$ in the cell is certainly to reduce the photolysis of NO$_2$ (Reaction R1); this can be achieved by a series of measures.

1. Only expose the cell to measurement radiation by, for example, putting it in a nontransparent tube.

2. Minimize exposure time by, for example, putting the cell in a light-tight box when not in use.

3. Use a filter in front of the cell which only admits radiation at wavelengths > 450 nm; this, however, may interfere with the measurements.

Also, it may be good to avoid ozone photolysis in the cell to minimize OH formation by using a UV-nontransparent cell material, e.g. glass instead of quartz. In addition, it is a good idea to keep the gas in the cell as dry as possible to avoid formation of HNO$_3$ or HNO$_2$ and to further minimize OH formation. Furthermore, it may be a good idea to illuminate a freshly filled cell initially (for a few hours) to allow all remaining water to be converted to HNO$_3$, thus (1) minimizing later changes in the NO$_2$ concentration due to HNO$_3$ formation and (2) avoiding HONO formation.

A further important measure is to add O$_2$ to the cell in order to enhance reconversion of any NO formed to NO$_2$.

The problems associated with excessive N$_2$O$_4$ formation in the cell (reduction of the NO$_2$ CD, temperature dependence of the NO$_2$ CD, and HNO$_3$ formation) can be reduced by using lower NO$_2$ concentrations in the cell. The length of the cell may need to be extended to still achieve a desired NO$_2$ CD. In principle the cell may also be heated to lower the amount of steady-state N$_2$O$_4$.

Problems with the optics of the cell are also difficult to avoid; fortunately they usually lead to changes in the NO$_2$ CD of < 10 %. In principle wedged cell windows or antireflective coatings could be used on the cell windows to minimize the problems described in Sect. 2. Another approach would be to tilt the entire cell with respect to the optical axis; thus reflected radiation would not reach the entrance optics of the spectrometer.

*Data availability.* The model data can be obtained from the authors upon request.

# Appendix A: Solution of the cubic equation for the stationary-state NO concentration

The above Eq. (7) is a cubic equation, which we recognize as Cardano's formula after substituting $z = [NO]$:

$$z^3 + pz + q = 0, \tag{A1}$$

for which the solution is well known as (Bronstein et al., 2013)

$$z = u + v, \tag{A2}$$

with

$$u = \sqrt[3]{-\frac{q}{2} + \sqrt{\Delta}}, \quad v = \sqrt[3]{-\frac{q}{2} - \sqrt{\Delta}}, \tag{A3}$$

and

$$\Delta = \left(\frac{q}{2}\right)^2 + \left(\frac{p}{3}\right)^3. \tag{A4}$$

Equation (7) for $z = [NO]$ thus becomes

$$[NO]^3 \cdot \frac{k_6}{2J_1} + [NO] - [NO_2]_0 = 0. \tag{A5}$$

It is transformed with $a = 2J_1/k_6 \approx 4 \times 8 \times 10^{-3}/1.95 \times 10^{-38} \approx 8.205 \times 10^{35}$ molec.$^2$ cm$^{-6}$ to

$$[NO]^3 + a\,[NO] - a\,[NO_2]_0 = 0. \tag{A6}$$

Sample solutions include the following.

1. 1000 hPa of initial $NO_2$, i.e. $[NO_2]_0 = 2.4 \times 10^{19}$, with $p = a$ and $q = -a[NO_2]_0 \approx -2.4 \times 10^{19} \cdot 8.205 \times 10^{35} \approx -1.969 \times 10^{55}$, and

$$\begin{aligned}\Delta &= \left(\frac{q}{2}\right)^2 + \left(\frac{p}{3}\right)^3 = \frac{a^2[NO_2]_0^2}{4} + \frac{a^3}{27} \\ &\approx 9.694 \times 10^{109} + 2.046 \times 10^{106} \\ &\approx 9.696 \times 10^{109}, \end{aligned} \tag{A7}$$

we obtain the only positive and real solution:

$$\begin{aligned}[NO] &= u + v = \sqrt[3]{-\frac{a[NO_2]_0}{2} + \sqrt{\Delta}} \\ &\quad + \sqrt[3]{-\frac{a[NO_2]_0}{2} - \sqrt{\Delta}} \\ &\approx \sqrt[3]{1.969 \times 10^{55}} + \sqrt[3]{-2 \times 10^{51}} \\ &\approx 2.700 \times 10^{18} - 1.26 \times 10^{17} \\ &\approx 2.574 \times 10^{18}. \end{aligned} \tag{A8}$$

This means that $[NO]/[NO_2]_0$ is about $2.574 \times 10^{18}/2.4 \times 10^{19} \approx 10.7\,\%$ of the initial $NO_2$.

2. At 100 hPa of initial $NO_2$ ($[NO_2]_0 = 2.4 \times 10^{18}$), we obtain $[NO]/[NO_2]_0 \approx 2.4 \times 10^{18} \approx 42.9\,\%$ of the initial $NO_2$.

3. At 10 hPa of initial $NO_2$ ($[NO_2]_0 = 2.4 \times 10^{17}$), we obtain $[NO]/[NO_2]_0 \approx 2.4 \times 10^{17} \approx 94.0\,\%$ of the initial $NO_2$.

4. At 1 hPa of initial $NO_2$, ($[NO_2]_0 = 2.4 \times 10^{16}$), we obtain $[NO]/[NO_2]_0 \approx 100\,\%$ of the initial $NO_2$.

# Appendix B: The simplified model

Reaction kinetic box-model calculations show the temporal evolution of $[NO_2]$, $[NO]$, and $[O_2]$, according to the simple reaction system (Reactions R1, R4, and R6) in an illuminated $NO_2$ cell. In addition we initially neglect the $NO_2$ dimer formation. These calculations merely serve to demonstrate that the analytical solution as derived in Sect. 3.1 matches the model calculations.

Figure B1 shows some results of this oversimplified model, assuming (as above) initial $NO_2$ levels $[NO_2]_0$ of 1, 10, 100, and 1000 hPa ($2.4 \times 10^{16}$, $2.4 \times 10^{17}$, $2.4 \times 10^{18}$, and $2.4 \times 10^{19}$ molec. cm$^{-3}$, respectively). As expected the initial $NO_2$ concentration drops within the first few seconds (at high initial $NO_2$) to minutes (at low $NO_2$) until the back reaction kicks in and leads to stationary-state levels of all species after this initial period. At 1 hPa of initial $NO_2$ its concentration drops to very small levels ($< 0.1\,\%$), as shown in Sect. 3.1, while at 1000 hPa we still see about a 0.7 % TS4 loss of initial $NO_2$. These figures are exactly the same as those found from the steady-state calculations (see Appendix A).

Figure B2 shows some results of the simplified model (Reactions R1, R4, and R6), but including the $NO_2$–$N_2O_4$ equilibrium (Reactions R7 and R8) for initial $NO_2$ levels, $[NO_2]_0$ of 1, 10, 71, 344 hPa ($2.4 \times 10^{16}$, $2.4 \times 10^{17}$, $1.7 \times 10^{18}$, $0.84 \times 10^{19}$ molec. cm$^{-3}$, due to filling the cell with $NO_2$ levels of 1, 10, 100, and 1000 hPa, respectively, which then immediately undergo $N_2O_4$ equilibration). For the lower initial $NO_2$ levels (1 and 10 hPa), there is little difference to Fig. B1. The initial $NO_2$ concentration drops within the first few seconds to minutes to small fractions of the initial $[NO_2]_0$. As discussed above (Sect. 3.3), the situation can be improved by adding initial $O_2$ (topped up to 1000 hPa). The thin blue line in the plots for $[NO_2]_0$ of 1, 10, and 71 hPa indicates the results for the corresponding $NO_2$ profiles. In particular at higher initial $NO_2$ levels (e.g. 71 hPa) the ultimate $NO_2$ levels are considerably enhanced by $O_2$ addition. However at higher initial $NO_2$ levels (see plots for 71 and 344 hPa initial $NO_2$) there is a large reduction in $NO_2$ due to the $NO_2$-dimer formation, inducing stronger temperature dependence.

In order to get a feeling for the influence of temperature changes in the model run for $[NO_2]_0 = 344$ hPa the temperature was raised by 5 K (298 to 303 K) after 100 s; the corresponding plot (bottom right in Fig. B2) shows an increase

in NO$_2$ (thin blue line) of about 16 % due to this temperature rise.

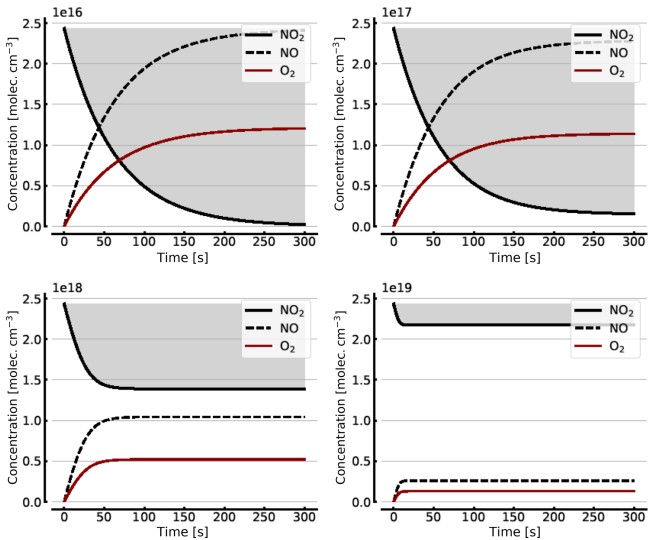

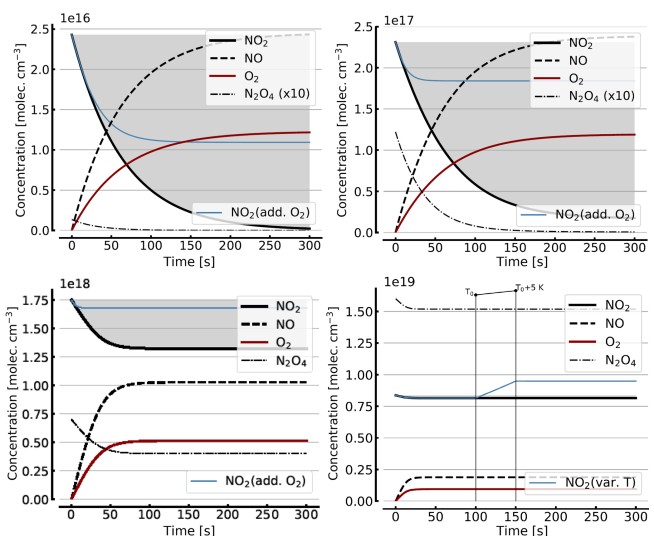

**Figure B1.** Model calculations of the temporal evolution of [NO$_2$] (thick solid black line), [NO] (dashed black line), and [O$_2$] (solid brown line) according to the simple reaction system (Reactions R1, R4, and R6 only, at 298 K) in an illuminated NO$_2$ cell. Here the NO$_2$–N$_2$O$_4$ chemical equilibrium is neglected, which makes in particular the plots for initial [NO$_2$]$_0$ = 1000 hPa unrealistic. All time series are for calculation with no added initial O$_2$. Initial [NO$_2$]$_0$ of 1, 10, 100, and 1000 hPa (2.4 × 10$^{16}$, 2.4 × 10$^{17}$, 2.4 × 10$^{18}$, and 2.4 × 10$^{19}$ molec. cm$^{-3}$, respectively).

**Figure B2.** Same model calculations as shown in Fig. B1, but including N$_2$O$_4$. Initial [NO$_2$]$_0$ of 1, 10, 71, and 344 hPa (2.4 × 10$^{16}$, 2.4 × 10$^{17}$, 1.7 × 10$^{18}$, and 0.8 × 10$^{19}$ molec. cm$^{-3}$, respectively; see text). Temporal evolution of [NO$_2$] (thick solid black line), [NO] (dashed black line), [O$_2$] (solid brown line), and [N$_2$O$_4$] (thin dashed–dotted line; ×10 in the upper two panels) according to the simple reaction system (Reactions R1, R4, R6, R7, and R8 only, at 298 K) in an illuminated NO$_2$ cell. All time series with the exception of the thin blue line (in the plots for [NO$_2$]$_0$ = 1, 10, and 71 hPa) are for calculation with no added initial O$_2$. The thin blue line (in the plots for [NO$_2$]$_0$ of 1, 10, and 71 hPa) indicates the evolution of NO$_2$ for a calculation with initial O$_2$ topped up to 1000 hPa. The plot for [NO$_2$]$_0$ = 344 hPa additionally shows the increase in NO$_2$ (thin blue line) at a temperature rise of 5 K (298 to 303 K).

*Author contributions.* Both authors developed the concept of the paper and discussed its contents. UP wrote most of the text, and JK performed the model calculations. CE3

*Competing interests.* The authors declare that they have no conflict of interest.

*Acknowledgements.* Partial support by the DFG project 193/18-3 is gratefully acknowledged. We also thank two anonymous reviewers for valuable and constructive comments.

*Financial support.* The article processing charges for this open-access publication were covered by the Max Planck Society.

*Review statement.* This paper was edited by Hendrik Fuchs and reviewed by two anonymous referees.

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

*Please note the remarks at the end of the manuscript.*

**Remarks from the language copy-editor**

CE1     Please confirm the edit made here.
CE2     Please confirm the edit made here.
CE3     Please confirm the minor edits here.

**Remarks from the typesetter**

TS1     Please double-check that all instances of [$NO_Z$] have been corrected.
TS2     The horizontal lines are according to our standards and cannot be changed.
TS3     Changing this value would need the editor's approval. Please send a document with an explanation of why this is necessary.
TS4     Changing this value would need the editor's approval. Please send a document with an explanation of why this is necessary.