# Peer review of "Caution with spectroscopic NO₂ reference cells (cuvettes)"

_Atmospheric Measurement Techniques, 2019_

## Short Comment (SC1) · 16 Apr 2019

Interesting paper on the reactions occurring in NO2 reference cells used in e.g. DOAS spectrometers. A few remarks:

1) Any comparison with experimental observations (either from the authors own experiments or from literature) is missing. It is suggested to add such a comparison (if these data are available).

2) NO2 cannot be obtained at high purity from commercial gas suppliers. Some comment could be added about this (i.e., starting mixture will already be more complex).

3) Page 11 "One can actually assume that all H2O is ultimately converted to HNO3, sequestering equivalent amounts of NO2 and water. "

This might be expected but apparently this does not happen. At VSL we did some experiments adding water to NO2 mixtures and only a relatively small part of the water is eventually converted to HNO3. (see https://www.hindawi.com/journals/jspec/2018/9845608/).

4) Topping with dry synthetic air is probably preferred over filling with laboratory air (p11).

5) In equation R20 on page 11 the value of the rate constant is missing.

6) The section on the path length of the optical cells (section 2) is not relevant for the rest of the paper and should be omitted here.

Stefan Persijn, VSL (Dutch Metrology Institute)
* * *

---

## Referee Comment (RC1) · Anonymous Referee #1 · 6 Jun 2019

This manuscript describes the limitations of the use of NO2 cuvettes for quantitative measurements. In the draft, the impacts of chemical processes occur in a NO2 cuvette are assessed via analytical calculations alongside box model simulations. Generally, the manuscript is well written and presents an interesting topic. I recommend a publication after the authors need to address the following questions/comments.

Comments:

1. Line 40-43 (page 1): The bullet points should follow the sequence of the draft.

2. Line 12-20 (page 1): Why the case related to NO2 is preferred? The selection needs some motivation.

3. Lines 11-16 (page 2): The use of anti-reflective coatings for windows of a cell is very

common. What will happen in the case of coated surfaces?

4. Line 29 (page 3): The statement "NO2 is a quite reactive gas" needs a justification (How?) or a reference.

5. Line 33 (page 3): I think the use of terminology should be 'accuracy' instead of 'precision'. Since the contributions from further reactions will also be a part of accuracy.

6. Line 9 (page 4): To validate the statement "However, this is a slow process", the reaction rate coefficient is required.

7. Line 22 (page 4): 'k6' unit is missing.

8. Lines 7-40 (page 5): What will be the impact/importance of the reaction 'R5' on the NO2 concentration?

9. Line 11 (page 11): How is it possible only for traces of water (but no other trace gases) to enter the cell? I think this line should be modified.

10. Line 21 (page 11): The value of 'k20' is missing.

11. Line 31 (page 11): The assumption "all H2O is ultimately converted to HNO3" needs a reference.

12. Line 35 (page 11): What is and why 'laboratory air'? Why not oxygen or synthetic-air? I think lines 34-37 need a realistic starting assumption.

13. Page 12: A separate column in 'Table 1' for references (instead of the superscript in column 1) will ease the reading process.

14. Page 14-15-16: The sub-panels of Fig. 4, Fig. 5, Fig. 6, and Fig. 7 should be labelled with the relevant pressure values. In Fig. 5, the scale for N2O4 (the top right and left panel) should be separate. For example, it can be done by plotting a separate y-axis (only for N2O4) on the right side.

15. Line 4 (Page 17): Duplicate pressure value needs to be removed.

16. Line 5-6 (Page 17): The statement 'initial O2 was assumed' contradicts 'Line 22 (Page 17) and Line 1 (Page 18)'. Which one is true?

17. Line 2-5 (Page 18): The statement "there are no fundamental differences in the NO2 time series between the simple model and the full model" is not understandable (what is referred?). A table, for the final NO2 concentration at a fixed time interval (@ 300 s), would be helpful to summarize the comparison (simple vs full model).

18. Line 29 (Page 18): The wavelength for the photolysis (threshold) of NO2 to NO conversion is required.

---

## Referee Comment (RC2) · Anonymous Referee #2 · 2 Jul 2019

In this submission, Platt and Kuhn describe how optics and chemistry alter the effect of reference cells, particularly the NO2 reference cell. Such cells are commonly used in DOAS and gas correlation spectroscopy for wavelength calibration and absolute calibration of column densities. Although the authors show that the influence of optical factors can be appreciable, the most important outcome of this work is that it draws attentions to the complexity of the chemical composition in the reference cell when exposed to light. Changes in chemical composition can be very large indeed, and inattention to these effects will compromise measurements.

The underlying reactions are well-known and relatively easily modelled and this work presents nothing new in this respect. From the practical point of view, such factors may not be widely appreciated. Therefore, from the instrument operator's perspective, the

authors offer a cautionary information about such cells, as well as practical suggestions to improve the stability of the chemical composition in these cells. The use of such cells and their influence on standard trace gas measurements by a widely used technique could make this work a valuable practical contribution to the literature. However, there are several aspects of this work that should be clarified and improved before publication can be recommended.

I found the manuscript unnecessarily convoluted and hard to read. A direct presentation of the calculations and simulation results would be clearer to the reader. In particular, the approach of gradually ramping up the complexity of the underlying reactions and analytical approximations seems unhelpful to this reviewer. Why not just present the full simulations? Limiting conditions could be examined separately.

1. Introduction

- The introduction is unusually short. The authors should explain how these reference cells are used in terms of calibration and column density, how this could lead to errors, and what sort of errors could arise. Would errors be expected for the wavelength calibration, for instance, even if the NO2 concentration in the cell changes? More equations in this part of the paper would be helpful.

- 44-45: Other cells and gases are mentioned; real-world examples should be provided. NO2 is relatively unusual as a strongly absorbing gas in the actinic region of the solar terrestrial spectrum.

2. Optics

- Section 2.1 is unclear whether it refers to a cell containing absorbing gas or not. This should be explicit, and the Beer-Lambert dependence on pathlength for the single and multiple passes should be clearly described.

- 16: The authors explain in the figure caption, but not the text, where the figure of 2% enhanced SCD comes from. It is unclear how this relates to the absorption coefficient

of the gas in the cell. In the case of strong absorption, the contribution of the multiple reflection becomes negligible even at relatively modest absorptions (e.g., exp (-a L) < 0.5 means the first multiple pass of three times through the cell is < 12% of the original value. Values for other column densities would be instructive here.

3. Chemistry

- p.3, 43: A J value is provided for R1 but it is unclear where this comes from. The formula describing the photolysis rate and relationship to the actinic flux, quantum yield, and absorption spectrum should be provided. There is no recognition in the text that this key parameter, which drives all the subsequent chemistry, can be highly variable. It depends on whether this is direct sunlight, and how it depends on season, latitude, time of day, etc. It is also unclear how this J could relates to values for an active DOAS system with an artificial light source, or even whether such cells are used in this case.

- The text does not describe whether the beam diameter completely or only partially fills the reference cell diameter. If the latter, then there is a more complex spatial dependence on the chemistry. Fortunately in this case, such a situation reduces the impact of the chemical reactions.

- The text should clarify that the value for k5 pertains to ambient pressures.

- Example 2 has limited value because it excludes R5 (which has approximately the same rate as R4). In the limit, reformed NO2 will eventually photolyse until NO is the sole product, but this process becomes increasingly slow. This is surely an important practical issue in the use of these cells.

- P6, 18: While the point made is correct, the time to attain equilibrium should be stated less precisely. The half-life is about 5 us.

- Fig. 3 may be simplified, but it seems odd to leave out these reactions: O + NO, O3 + NO2, and 2NO2 + H2O –> HONO + HNO3

- P.10, 35: State that this is a slow reaction in the gas phase

- P11,21: "??" ??

- 2NO2 + H2O –> HONO + HNO3 is an important heterogeneous reaction and would certainly be expected to occur in reference cells, including under dry conditions as it is very difficult to get a completely dry well. This should be included in the simulations.

4. Simulations

- I find little value in Figs 4 & 5 which simulate simplified chemistry. What's the point? Simplifying cases of the full chemical system can be discussed in the text.

- In contrast, Fig.6 could be split into several figures for presentation clarity.

- The effect of changing J values was not investigated, though this would strongly influence the chemistry in the cells.

5. Summary and conclusions

Another possibility is to use wedged windows for the cell (which would halve the reflections assuming internal window are parallel) or parallel windows angled sufficiently to avoid multiple passes.

Minor comments:

Abstract:

21: "at" → "using" / "for" etc. Awkward sentence.

25: particularly

p.3, 4: "oft" → "of"

p3, 23: sentence fragment needs clarification

---

## Author Comment (AC1) · 31 Aug 2019

**The authors' answers to interactive comments on "Caution with Spectroscopic NO2 Reference Cells (Cuvettes)" by Ulrich Platt and Jonas Kuhn**

First of all we like to thank the anonymous reviewer for the constructive and helpful comments. We substantially revised our manuscript in the light of these comments and we are confident that we addressed all points, which were raised. We trust that our manuscript is significantly improved now and will be ready for publication in AMT. We also thank the reviewers in the acknowledgement.
In the following we reproduce the reviewers' comments in red and add our answers (plus in most cases description of changes to the manuscript) in black.

**Anonymous Referee #1**

This manuscript describes the limitations of the use of NO2 cuvettes for quantitative measurements. In the draft, the impacts of chemical processes occur in a NO2 cuvette are assessed via analytical calculations alongside box model simulations. Generally, the manuscript is well written and presents an interesting topic. I recommend a publication after the authors need to address the following questions/comments.

Comments:

1. Line 40-43 (page 1): The bullet points should follow the sequence of the draft.

Response: We re-ordered the bullet points.

2. Line 12-20 (page 1): Why the case related to NO2 is preferred? The selection needs some motivation.
Response: We modified the first sentence of the $2^{nd}$ paragraph of the abstract and inserted a motivating sentence. It now reads: 'Since NO2 is a particularly popular molecule to be studied by spectroscopic measurement techniques in the atmosphere (e.g. DOAS) and at the same time can be unstable in cells we chose it as an example to demonstrate that the effective CD seen by the instrument can deviate greatly (by orders of magnitude) from expected values.'

3. Lines 11-16 (page 2): The use of anti-reflective coatings for windows of a cell is very common. What will happen in the case of coated surfaces?
Response: We doubt somewhat that the use of anti-reflective coatings for windows of cells used for verification of DOAS instruments is very common. Nevertheless, this would be an approach to reduce the described effect of multiple passes. The consequence of reduced reflection can be readily calculated from the equation in the text following Eq. (1) by inserting smaller numbers for R. For instance assuming R=0.01 instead of 0.04 would reduce the fraction of radiation passing the cell three times to about 0.04%.
We added the comment '(this effect could be reduced by adding anti-reflective coatings to the cell windows)' after the sentence ending with 'three times'.

4. Line 29 (page 3): The statement "NO2 is a quite reactive gas" needs a justification (How?) or a reference.

Response: The bulk of the manuscript deals with the many reactions of NO2, so it must be quite reactive. We, nevertheless, add the phrase '(see rest of the section)' to the revised version of the manuscript in order to make it clear why we come to this conclusion.

5. Line 33 (page 3): I think the use of terminology should be 'accuracy' instead of 'precision'. Since the contributions from further reactions will also be a part of accuracy.

Response: We agree with the reviewer and changed the term accordingly.

6. Line 9 (page 4): To validate the statement "However, this is a slow process", the reaction rate coefficient is required.

Response: We agree, the reaction rate is given in Table 1 (Reaction No. 2). Of course the process is mainly slow because the concentration of O-atoms is very low ($<10^9$ cm$^{-3}$, see Figure 6).
We, therefore, added the value of the reaction rate constant to the revised version of the manuscript and added a comment to this effect: '… because the O-atom concentration will be very low (see Figures 4 to 8)'

7. Line 22 (page 4): 'k6' unit is missing.

Response: We apologize for this omission and added the units (cm$^6$molec$^{-2}$s$^{-1}$).

8. Lines 7-40 (page 5): What will be the impact/importance of the reaction 'R5' on the NO2 concentration?

Response: Due to the low O-atom concentration (see above) the impact of R5 will usually be very minor. The only exception is a scenario when no O2 is added to the cell and the NOx level is very low.

9. Line 11 (page 11): How is it possible only for traces of water (but no other trace gases) to enter the cell? I think this line should be modified.

Response: Water is (1) by far the most abundant trace gas (mixing ratios around 1% in ambient air), also water is always attached to the walls of a cell (typically one to several layers of H2O). Thus it is clear that H2O usually will be the most likely contamination.
We modified the sentence (as suggested by the reviewer) to read: 'Since water is by far the most abundant (typical mixing ratios around 1%) reactive trace gas in the ambient atmosphere (not counting noble gases, CO2, H2, or N2O) it may be possible

that traces of water enter the cell when it is filled, then a series of additional reactions may play a role:'

**10. Line 21 (page 11): The value of 'k20' is missing.**

Response: We apologize for this omission and added the value of k20 (note that the value is listed in Table 1).

**11. Line 31 (page 11): The assumption "all H2O is ultimately converted to HNO3" needs a reference.**

Response: This is not an assumption but the consequence of the reaction mechanism in the cell. Nitric acid is formed from NOx plus H2O (R19, R23) and since the amount of NOx by far exceeds the amount of H2O the reaction will ultimately consume all the H2O. This can also be seen in Figures 4-8.
See also answer to Stefan Persijn, point 3).

**12. Line 35 (page 11): What is and why 'laboratory air'? Why not oxygen or synthetic air? I think lines 34-37 need a realistic starting assumption.**

Response: After a cell is filled with NO2 at low pressure an obvious idea is to just open the valve and let ambient air (usually laboratory air) rush in. Thus, air with the typical humidity of ambient air (e.g. 70% at 25 degrees Celsius as stated in line 35) will enter the cell. Clearly, it is important to keep the amount of water in the cell to a minimum. Dry synthetic air, or better, dry oxygen are preferred as stated in section 5.2.
To make this clearer we changed the text in the revised version to read: '… is topped with ambient (e.g. laboratory) air (which is of course not recommended, see section 5.2) …'

**13. Page 12: A separate column in 'Table 1' for references (instead of the superscript in column 1) will ease the reading process.**

Response: We agree in principle, however adding a separate column to Table 1 would make the table too wide for vertical display. Therefore we prefer to leave Table 1 as it is.

**14. Page 14-15-16: The sub-panels of Fig. 4, Fig. 5, Fig. 6, and Fig. 7 should be labelled with the relevant pressure values. In Fig. 5, the scale for N2O4 (the top right and left panel) should be separate. For example, it can be done by plotting a separate y-axis (only for N2O4) on the right side.**

Response: This is a good idea, since we split Fig. 6 into new Figures 4-8 (as suggested by reviewer #2) we added the values for the initial NO2 partial pressure in the individual figure captions. In Fig. 5 (now Figure 11 in appendix 2) we plotted [N2O4]*10 in the first two plots (for [NO2]0 = 1, 10 hPa).

**15. Line 4 (Page 17): Duplicate pressure value needs to be removed.**

Response: We apologize for this error and changed the caption of Figure 6 (now split up into Figures 4-8 as suggested by reviewer #2).

**16. Line 5-6 (Page 17): The statement 'initial O2 was assumed' contradicts 'Line 22 (Page 17) and Line 1 (Page 18)'. Which one is true?**
Response: We can not find a contradiction here, in line 5 (page 17 of the initial version) we state 'Except for the top panel initial O2 was assumed.' while line 22 reads 'In the top panel no initial O2 was assumed …'.

**17. Line 2-5 (Page 18): The statement "there are no fundamental differences in the NO2 time series between the simple model and the full model" is not understandable (what is referred?). A table, for the final NO2 concentration at a fixed time interval (@ 300 s), would be helpful to summarize the comparison (simple vs full model).**

Response: We find that the statement in lines 2-5 of page 18 is quite straightforward. Nevertheless, we added Table 2 comparing the NO2 concentrations at 1000 seconds for the various model runs.

**18. Line 29 (Page 18): The wavelength for the photolysis (threshold) of NO2 to NO conversion is required.**

Response: The NO2 photodissociation threshold wavelength is 398 nm (e.g. Johnston and Graham, 1974, Burkholder et al., 2015), the quantum yield for photolysis sharply drops above this threshold but is reaches zero only at about 430 nm, probably since vibrational excitation of the NO2 molecule supplies the energy deficit at wavelengths exceeding 398nm (See e.g. Mérienne et al., J Atmos Chem (1995) 20: 281. https://doi.org/10.1007/BF00694498). This is already stated in the text following Reaction 1 near the bottom of page 3 (original manuscript).
So no change to the manuscript is required here, we just used the opportunity to slightly change these numbers to the more exact values 398nm (threshold) and 430nm (quantum yield virtually zero at this and any longer wavelength), respectively in the revised version of the manuscript.

---

## Author Comment (AC3)

**Answer to Interactive comment By Stefan Persijn,** VSL (Dutch Metrology Institute), spersijn@vsl.nl,
**on**
**Atmos. Meas. Tech. Discuss., doi:10.5194/amt-2019-130, 2019, entitled:**
**"Caution with Spectroscopic NO$_2$ Reference Cells (Cuvettes)" by Ulrich Platt and Jonas Kuhn**.

Answer:
We like to thank Stefan Persijn for his interesting and constructive comments, which we answer in the following. The comments are reproduced in slant font followed by our answers in normal font.

*Interesting paper on the reactions occurring in NO2 reference cells used in e.g. DOAS spectrometers.*
*A few remarks:*

*1) Any comparison with experimental observations (either from the authors own experiments or from literature) is missing. It is suggested to add such a comparison (if these data are available).*

Answer: This is a theoretical study, meaningful experimental data from cuvettes do not appear to be available in the literature. Special measurements to 'validate' NO$_X$ reaction system, which extremely well studied in the laboratory (see reaction kinetic data compilation JPL 15-10, Burkholder et al. 2015 as referenced in our manuscript) appear to be a waste of time.

*2) NO2 cannot be obtained at high purity from commercial gas suppliers. Some comment could be added about this (i.e., starting mixture will already be more complex).*

We like to thank you for this comment. Although neither the basis for this statement nor any quantitative information is given, it appears to be plausible and we shall add a comment in the revised version of the manuscript saying that 'In fact, when buying NO$_2$ from a manufacturer some of the described reactions can already proceed in the initial gas, which therefore might already contain impurities (e.g. of NO, HONO and HNO$_3$)'.

*3) Page 11 "One can actually assume that all H2O is ultimately converted to HNO3, sequestering equivalent amounts of NO2 and water. " This might be expected but apparently this does not happen. At VSL we did some experiments adding water to NO2 mixtures and only a relatively small part of the water is eventually converted to HNO3. (see https://www.hindawi.com/journals/jspec/2018/9845608/).*

Answer: In the quoted publication (S. Persijn, Purity Analysis of Gases Used in the Preparation of Reference Gas Standards Using a Versatile OPO-Based CRDS Spectrometer, J. of Spectroscopy, Vol. 2018, Article ID 9845608) some experiments

are described, where 2 ppm water (vapour) were added, leading to the formation of gas phase $HNO_3$ amounting to between 2 and 25% of the added water. Since it is likely that a good fraction of $HNO_3$ formed will stay at the walls of the vessel (this problem is also pointed out in the publication), these figures have to be regarded as lower limits of the $H_2O$ to $HNO_3$ conversion. Moreover, the experiments were performed at very low (for absorption cells) $NO_2$ mixing ratios of only 10 ppm, thus $N_2O_4$ formation should be negligible. Also the time between water addition and $HNO_3$ measurement is not given. Therefore we can not see the evidence for the statement that $H_2O$ may not be quantitatively converted to $HNO_3$ in an environment containing very high (e.g. thousands of ppm's) $NO_X$ levels.

In fact our model calculations (see Figures 4 to 8 of the revised version) that water is quickly lost in the cell.

*4) Topping with dry synthetic air is probably preferred over filling with laboratory air (p11).*

Answer: We actually write in section 5.2 that we recommend topping with dry air or oxygen. Whether synthetic air is sufficiently dry is a good question. We would prefer air (or better oxygen as pointed out e.g. in the Examples on page 7 and section 5.2), which is dried by a cartridge with drying agent (e.g. molecular sieve) in a cartridge.

*5) In equation R20 on page 11 the value of the rate constant is missing.*

Answer: Thank you for pointing out this omission, which we will correct in the revised version. Note, however, that the value of $k_{20}$ (and its temperature dependence) is given in Table 1 of our manuscript.

*6) The section on the path length of the optical cells (section 2) is not relevant for the rest of the paper and should be omitted here.*

Answer: We disagree with this statement. Our manuscript is about potential problems with $NO_2$-cells not only about chemistry in cuvettes. We therefore, find it natural and necessary to also report on other effects influencing the apparent optical density of a cuvette.

---

## Author Response (AR1)

Dear Editor,

We just uploaded the revised version of our manuscript amt-2019-130, entitled:
"Caution with Spectroscopic NO2 Reference Cells (Cuvettes)"
by U. Platt and J. Kuhn.

We already uploaded comprehensive answers to the reviewers' comments complete with descriptions of the changes to the manuscript.

In addition Winword files of the manuscript with the changes marked are included in a *.zip file.

We trust that on this basis you can make a positive decision with regard to publishing our manuscript.

With very best regards
Ulrich Platt